# GraphTranslator: Aligning Graph Model to Large Language Model for Open-ended Tasks

## ABSTRACT

Large language models (LLMs) like ChatGPT, exhibit powerful zero-shot and instruction-following capabilities, have catalyzed a revolutionary transformation across diverse research fields of artificial intelligence especially for open-ended tasks. While the idea is less explored in the graph domain, despite the availability of numerous powerful graph models (GMs), they are restricted to tasks in a pre-defined form. Although several methods applying LLMs to graph have been proposed, they fail to simultaneously handle the pre-defined and open-ended tasks, by using LLM as node feature enhancer or as a standalone predictor. To break this dilemma, we propose to bridge the pretrained GM and LLM by a Translator, named *GraphTranslator*, aiming to leverage GM to handle the pre-defined tasks effectively and utilize the extended interface of LLMs to offer various open-ended tasks for GM. To train such Translator, we propose a Producer capable of constructing the graph-language alignment data along node information, neighbor information and model bias. By treating the node representation as a type of language, the proposed *GraphTranslator* empowers a LLM to make predictions based on node representation and language instructions, providing a unified perspective for both pre-defined and open-ended tasks. Extensive results show that the proposed *GraphTranslator* effectively improves the results of zero-shot node classification. The preliminary graph question answering experiments reveal our *GraphTranslator* potential across a broad spectrum of open-ended applications through language instructions.

## CCS CONCEPTS

• **Computer systems organization** → **Embedded systems**; • **Networks** → Network reliability.

**ACM Reference Format:**
Anonymous Author(s). 2018. GraphTranslator: Aligning Graph Model to Large Language Model for Open-ended Tasks. In *Proceedings of Make sure to enter the correct conference title from your rights confirmation emai (Conference acronym 'XX)*. ACM, New York, NY, USA, 13 pages. https://doi.org/XXXXXXX.XXXXXXX

## 1 INTRODUCTION

Graph is commonly used to model many real-world relationships, such as social networks [23], citation networks [15] and the e-commerce networks [42]. In recent years, graph models (GMs),

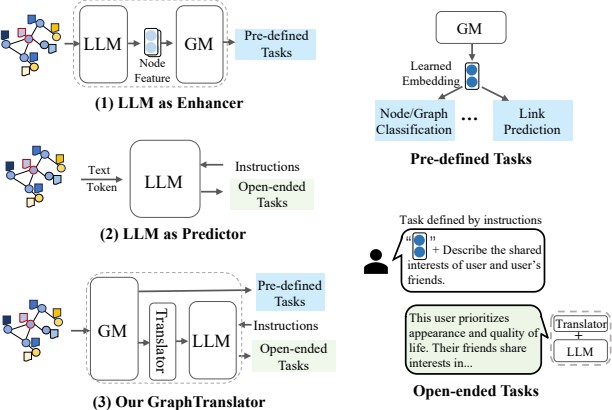

**(a) Existing works versus our work**

**(b) Pre-defined and open-ended tasks in our work**

**Figure 1: Intuitive illustration of the proposed *GraphTranslator* from two perspectives: (a) Comparisons of *GraphTranslator* with popular paradigms of applying LLMs to Graph. Unlike using LLM as enhancer or sole predictor, *GraphTranslator* bridges LLM and GM, handling both pre-defined and open-ended tasks simultaneously. (b) Simple demonstration of tasks in *GraphTranslator*, where GM is leveraged for pre-defined tasks, and the LLM is extended as the interface of node embeddings learned by GM, enabling to make prediction for tasks defined by instructions.**

such as graph neural networks (GNNs) [9, 15], which combine node feature information with the graph structure by using neural networks, have achieved state-of-the-art performance on a wide range of real-world applications. Despite great success, GMs are restricted to tasks within pre-defined format (e.g., node classification). They only identify pre-defined classes that are present in training phase, which inevitably makes it challenging for GMs to generalize to unseen categories and concepts.

Recently, the emergence of large language models (LLMs) like ChatGPT[1] has brought about a paradigm shift in natural language processing (NLP) research, showcasing their impressive emergent abilities [31] for flexible open-ended tasks based on natural language instructions. Such development of LLMs also has revolutionized research of diverse modality [16, 17, 33]. Taking image as an example, LLMs largely facilitates the visual-centric open-ended tasks, such as instructed image-to-text generation and visual question answering tasks, transforming how we interpret and interact with visual information.

---

[1] https://openai.com/blog/chatgpt

Similarly, for graphs, beyond the foundational pre-defined tasks, there is a strong need for empowering the open-ended tasks. Especially for the text-attributed graphs, which are commonly used in social media and e-commercial, supporting tasks that can be customized by users with text instructions and yield interpretable responses, will greatly enhance the user experience and expand the business scope. To this end, several works have applied LLMs for graph recently, which can be categorized into two classes [3] as showed in Figure 1 (a)-1 and (a)-2: first, leveraging LLMs to enhance nodes' text attributes with their massive knowledge and then generating predictions through GMs [3, 4, 6, 11, 34]; second, regarding node as token or text then employing LLM as standalone predictor [3, 8, 28, 37]. However, when it comes to practical industrial scenarios, there is a dilemma: The former methods, using LLM as the enhancer of GM predictor, can produce accurate prediction on pre-defined tasks, but fails to process open-ended tasks and lacks interactivity and explainability. The latter methods, employing a LLM as sole predictor, can handle the open-ended tasks while may bring the hallucinations [39], low speed and high cost of LLMs, which is unbearable for pre-defined tasks. It naturally raises a question: *Can we build a model that can solve both pre-defined and open-ended tasks?*

To answer this question, we propose to align the pre-trained GM to LLM, where GM focuses on the pre-defined tasks, and LLM serves as a interface of GM for open-ended tasks. However, it is non-trivial to align GM to LLM for facing two challenges: (1) There exists a significant modality gap between the trained GM and LLM, due to their differences in data format and processing mechanisms. LLMs only operate on sequences of tokens representing natural language text, and they are trained to understand and generate human-readable text. While Graph models process structured graph data and output node embeddings. These embeddings capture the structure and features of graph but are not inherently interpretable as natural language. (2) There lacks alignment data for bridging GM and LLM, which is crucial to leverage their individual strengths in an integrated system. Without natural alignment data, it's difficult to train models to understand and translate between the two modalities (node embeddings and textual tokens) effectively. Directly converting may result in loss of information or introduce noise. Ideally, for seamless alignment, there should be a dataset with pairs of node embeddings and corresponding textual descriptions, allowing the model to learn the intricate alignment between GM and LLM.

In this paper, to address above challenges, we propose a novel framework called *GraphTranslator* to align the pre-trained GM to LLM, solving both pre-defined and open-ended tasks. Specifically, for the challenge of modality gap, *GraphTranslator* introduces a Translator module which to converts node embedding into token embedding space, which can be interpreted by LLM. To achieve this, the Translator module learns a set of graph queries to extract the language information of node embeddings, then performs generative learning for adapting to LLM. For the second challenge of lacking alignment data, we introduce a Producer that capable of constructing (node embedding, textual description) pairs through the powerful generation ability of LLMs. To seamlessly textualize the information encoded in node embeddings, we generate the description step by step, including briefing node's attribute, summarizing neighbor attribute, then reasoning their commonality. After

training on the alignment data, as presented in Figure 1 (b), a frozen LLM, equipped with Translator, can handle various open-ended tasks based on the node embedding and instructions. In conclusion, in our *GraphTranslator* , pre-defined tasks can be tackled efficiently by the customized GM, while LLMs further provide GM with interactivity and interpretability.

We highlight our contributions as follows:

- We propose a novel model *GraphTranslator* that aligns graph model to large language model, providing a unified perspective for both pre-defined and open-ended tasks.
- *GraphTranslator* introduces a Translator module to eliminate the modality gap, by converting node embeddings learned by GM to a set of tokens. For further training, a Producer module is designed to generate the alignment data, through seamlessly textualizing the information encoded in node embeddings.
- We evaluate our method on real-world datasets for the open-ended tasks. The experimental results demonstrate the effectiveness of *GraphTranslator* on zero-shot node classification. The preliminary graph question answering experiments reveal the noteworthy potential of *GraphTranslator* when applied to tasks predicated upon language instructions.

## 2 METHODOLOGY

In this section, we first present the notations and problem settings used in our model named *GraphTranslator*, then introduce the architecture of *GraphTranslator* along with the training strategies.

### 2.1 Notations and Problem Settings

**Text-Attributed Graphs**    We mainly focus on the ubiquitous text-attributed graphs (TAGs), where nodes represent textual entities such as documents or sentences, and edges denote the relationships between them [11]. The representation learning on TAGs has attracted attention for past years and is applied to broad applications, ranging from text classification [12] to fake news detection [19]. Formally, we define a TAG as $\mathcal{G} = (\mathcal{V}, A, \{s_v\}_{v \in \mathcal{V}})$, where $\mathcal{V}$ is a set of $N$ nodes, and $A \in \{0, 1\}^{N \times N}$ is the adjacency matrix of graph. For each node $v$, it is associated with a sequential text feature, denoted as $s_v$. Here we use a subset of $N_P$ nodes for training *GraphTranslator*, denoted as $\mathcal{V}_P \subset \mathcal{V}$.

**Pre-defined Tasks**    In the current landscape of the graph domain, numerous graph models are mainly designed and trained for pre-defined tasks, which refer to tasks that are explicitly defined and specified in advance. These tasks typically have well-defined input and output specifications, along with clear evaluation metrics. When training graph models, researchers or engineers will define these tasks in advance and provide datasets associated with them to train the models. This allows models to focus on solving specific problems and achieve high performance on these tasks, such as node/graph classification[35], link prediction[38, 41], node clustering, etc. On the one hand, these well-formalized tasks provide a benchmark for model evaluation, on the other hand, these tasks often serve as the core function of real-world graph systems, requiring high levels of efficiency and precision, such as daily update in e-commerce system.

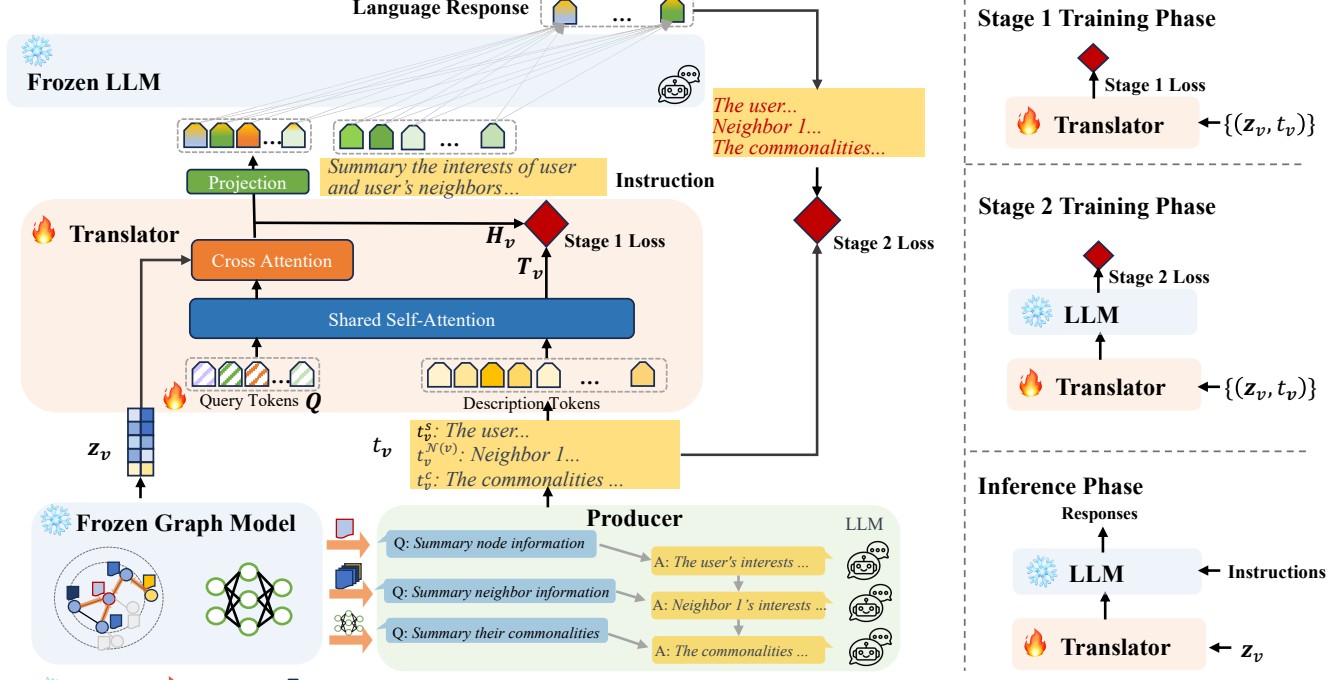

**Figure 2: The overall framework of the proposed *GraphTranslator*, which aligns GM to LLM by Translator for open-ended tasks. We train the lightweight Translator module following a two-stage joint training paradigm, with the alignment data generated by our Producer.**

**Open-ended Tasks** On the contrary, open-ended tasks offer greater flexibility and freedom, characterized by the absence of explicit task specifications or evaluation criteria. Models designed for open-ended tasks often depend on autonomous learning and creative problem-solving approaches. In real-world scenarios, new tasks often emerge with evolving business requirements, such as classifying new labels or tasks driven entirely by human instructions. The computer vision community has also adopted the language instruction paradigm for tasks like image-to-text generation [16, 17]. However, current graph models are constrained by predefined tasks and lack the flexibility to accommodate open-ended task customization guided by language instructions like LLMs.

## 2.2 Overall Architecture

The primary goal of our proposed *GraphTranslator* is to align graph models to LLMs, in order to harness the emergent capabilities of LLMs for open-ended tasks. Specifically, *GraphTranslator* consists of four components: (1) Frozen graph model (GM), is pre-trained on a large-scale graph, such as e-commerce graphs with billions of nodes, yielding embeddings for all nodes that encoding the graph's information for downstream tasks. We use the pre-trained Graph-SAGE model as an example in this work. (2) Frozen LLM, is trained on broad text corpus, showcasing emergent abilities when the number of parameters reach a certain scale. We employ the pre-trained

ChatGLM2-6B for demonstration. (3) Producer, is designed to construct the alignment data for training the Translator module, i.e., (node embedding, textual description) pairs. (4) Translator module, is designed to project the GM learned node embeddings into LLM space, eliminating the modality gap between GM and LLM.

Figure 2 illustrates the pipeline of our *GraphTranslator*. With the node embedding from pretrained graph model, the Producer first textualizes target node, neighbors and their commonality, for constructing alignment pairs. In first-stage training, the Translator is trained with the (node embedding, text description) pairs for alignment. In order to facilitate the node embedding to follow instructions better, we bridge the Translator with LLM, then fine-tuning the Translator with the pairs. To the end, this framework can be generalized to unseen node representation in inference phase, solving open-ended tasks by conversation.

## 2.3 Frozen Graph Model

The representation learning of TAGs has been extensively studied. Given a TAG $\mathcal{G} = (\mathcal{V}, A, \{s_v\}_{v \in \mathcal{V}})$, the typical graph neural networks (GNNs) [9, 15, 27] is denoted as $g_\theta(A, X)$, where $\theta$ is set of learnable parameters, $X$ is the node features processed by shallow methods such as bag-of-words (BoW) [10] or skip-gram [21]. Taking the GraphSAGE [9] as an example, typically, GraphSAGE samples a fixed-size neighbors $\mathcal{N}(v)$ around target node $v$, then concatenate the node's previous layer embedding $h_v^{k-1}$ with the

aggregated neighborhood vectors $\{h_u^{k-1}, \forall u \in \mathcal{N}(v)\}$ by:

$$h_v^k = \sigma(W^k \cdot \text{CONCAT}(h_v^{k-1} \cup \text{AGGREGATE}_k\{h_u^{k-1}, \forall u \in \mathcal{N}(v)\})). \quad (1)$$

Finally, the pre-trained GM $g_{\theta^*}$ encodes the local graph information of $v$ and yields node embedding $z_v = g_{\theta^*}(A, X)_v$.

Like GraphSAGE, the majority of GNNs can be viewed as a specific type of low-pass filtering, where node embeddings are smoothed through neighbor aggregation. This inherent model bias emphasizes the shared characteristics of node features while overlooking their differences. As a result, the representations of connected nodes tend to converge and become more alike, as noted in [2].

## 2.4 Frozen Large Language Model

LLMs are trained on extensive text corpora, acquiring a substantial amount of knowledge. It's worth noting that LLMs reveal their capabilities only when they reach a certain parameter scale [30]. To prevent the potential issues of catastrophic forgetting and the unbearable training costs associated with handling a large number of parameters, here we keep the LLM parameters fixed. Specifically, we employ ChatGLM2-6B, which is open-source bilingual (Chinese-English) language model. ChatGLM2-6B employs a specific type of autoregressive blank infilling task, which aligns with the typical design philosophy of most pretraining tasks. This approach involves a "disrupt and reconstruct" strategy, wherein portions of the original text are masked (disrupted) and subsequently predicted (reconstructed). After extensive training on large-scale corpora, these LLMs acquires the ability to retain a considerable amount of knowledge and provide reasonable answers to human queries.

## 2.5 Producer Module

To align GM and LLM, we construct the alignment data $P = \{z_v, t_v\}_{i=1}^{N_P}$, where $t_v$ outline the information encoded within node embedding $z_v$ for each node $v \in \mathcal{V}_P$. This process is not only related to graph data but also intertwined with the design of the GMs, so we employ LLM to construct high-quality description text with Chain-of-Thought (COT). Taking GraphSAGE as an example, we have devised a pipeline that guides LLM in constructing descriptive information along three key dimensions:

• Node Information: Node information of node is highly significant and is preserved in node embedding. Generally, node attributes, such as text or numerical data, are considered as one of the features for each node. These attributes are transformed into node features, which is achieved by Bag-of-Word or word embeddings models. Therefore, the Producer uses LLM to summarize and analyze the attributes of each node $v$ in the training set, yielding the self information description, denoted as $t_v^s$.

• Neighbor Information: GraphSAGE also consider neighboring information. GraphSAGE randomly samples a subset of neighboring nodes $\mathcal{N}(v)$ and aggregate their representations, yielding the neighbor embedding. Node self and neighbor information are further fused through weighted summation or concatenation. The Producer proceeds to employ LLM to summarize the attributes of the sampled neighbors $\mathcal{N}(v)$, resulting in the neighbor information description, denoted as $t_v^{\mathcal{N}(v)}$.

• Model Bias: Given that GraphSAGE operates as a low-pass filter,

tending to uncover similarities between nodes and their neighbors for smoothing purposes, we instruct LLM to consolidate shared information. Therefore, the Producer further utilizes LLM to infer the commonalities between node $v$ and its neighbors $\mathcal{N}(v)$ based on $t_v^s$ and $t_v^{\mathcal{N}(v)}$, resulting in commonality information denoted as $t_v^c$.

Through this carefully designed pipeline, we guide LLM step by step to construct high-quality embedding description text $t_v$ for each node $v \in \mathcal{V}_P$, by concatenating node self information, neighbor information, and the model bias, termed $t_v = \{t_v^s, t_v^{\mathcal{N}(v)}, t_v^c\}$.

## 2.6 Translator Module

There exists modality gap between the trained GM and LLM, LLMs fail to interpret node representations. Namely, the sizes of the node embedding and the input token of LLM are different, and they have different feature space. To resolve this discrepancy, we introduce the Translator module, which aims to align GM and LLM by converting the learned node embedding into token representations. A naive solution is applying a simple trainable projection matrix can convert $z_v$ into language embedding tokens, aligning their dimensionality with that of the word embedding space within the language model. While the simple transformation is hard to extract and translate the complex information contained within node representations to natural language, and may struggle to generalize to unseen nodes.

In our Translator module, for a pair $(z_v, t_v)$ in alignment data, we utilize two encoders, denoted $f_z(\cdot)$ and $f_t(\cdot)$, to extract their language features for alignment. For textual description $t_v$, we leverage the text encoder $f_t(t_v)$ (e.g., BERT [5]) to extract the language features $T_v = f_t(t_v)$, where $f_t(\cdot)$ contains 12 layers of Transformer blocks. For node embedding $z_v$, we also adopt a transformer-based network $f_z(\cdot)$ with $M$ learnable token embeddings as input, termed query tokens $Q = \{q_i\}_{i=1}^M$, and output $M$ features $H_v = \{h_{v,i}\}_{i=1}^M$ and $H_v = f_z(Q, z_v)$, extracting the information of $z_v$ that is most related to $t_v$. To achieve this, as inspired by [16], the query tokens $Q$ are designed to interact with each another using self-attention layers, interface with node embedding $z_v$ through cross-attention layers, and communicate with description $t_v$ by harmonizing the self-attention layers between $f_t$ and $f_v$.

## 2.7 Model Training

We train the lightweight Translator module following a two-stage joint training paradigm, bridging the gap between graph and LLMs step by step. In the first stage, we train the Translator module for extracting $H_v$ from the node embedding $z_v$ most relevant to $t_v$. In the second stage, we perform the generative learning by connecting the output of the Translator to the frozen LLM. We continue to training the Translator such that its output can be understood by the LLM.

**Stage 1**: Training the Translator for GM-text alignment. In training, we keep the pretrained node representations frozen and only train Translator module. We jointly optimize three objectives to align $H_v$ and $\tilde{t}_v$, which is the [CLS] token embedding of $T_v$. First, the contrastive objective aligns $H_v$ and $\tilde{t}_v$ by maximizing their mutual information. We first compute the pairwise similarity between $\tilde{t}_v$ and each token in $H_v$, and select the highest one as the similarity score, then contrast the similarity of a positive pair against

those of negative pairs. Second, the generative objective aims to train the Translator module for generating text based on the given embedding $z_v$. The essential information of given $z_v$ is extracted by query tokens $Q = \{q_i\}_{i=1}^{M}$ in $f_z$, then is seamlessly relayed to text tokens in $f_v$ through the shared self-attention layers. Now we replace the [CLS] token with [DEC] token for the generation task. By optimizing the cross entropy loss between the generated text and the actual description $t_v$, the $Q$ is forced to capture more details in $z_v$ related to the $t_v$. Third, the matching objective aims to learn the fine-grained alignment. We concatenate each token $h_{v,i} \in H_v$ ($i \in [1 \cdots M]$) with the [CLS] token $\tilde{t}_v$ of $T_v$, then feed them into a binary classifier and compute the matching score by averaging the logits across all queries.

**Stage 2**: Training the Translator for GM-LLM Alignment. We use a linear layer to project the output of Translator module, i.e., token embeddings $H_v$, into the same dimension with the word embedding of LLM. And the projected embeddings, which can be regarded as a soft prompt, are concatenated with human instructions as the input of LLM. Then we perform generative learning to tune the parameters of Translator with alignment data. In this way, the node embedding $z_v$ can be aligned with the pre-trained LLM word embedding.

# 3 EXPERIMENT

## 3.1 Experimental Setting

*3.1.1 Dataset.* We conducted experiments of the proposed *Graph-Translator* on real-world datasets, including the industrial dataset Taobao and the widely used benchmark dataset ArXiv:

• Taobao dataset is a subset extracted from the Taobao e-commerce platform. It consists of 980,000 nodes representing unique Taobao users. The associated attributes include user behaviors like purchases, searches, browsing, favorites, and cart additions. The extracted graph includes 1,790,000 edges, indicating social connections between users.

• ArXiv dataset is a graph constructed by a collection of ArXiv research papers with 169,343 nodes and 1,166,243 edges. Each node represents an individual research paper with textual attributes including the paper's title and abstract, and the edges reflect the paper citation relationship. For training the graph model, we divide nodes into 90,941 training nodes and 29,799 validation nodes following [13]. In consideration of ChatGLM2-6B's inference speed, our test set contains 4,000 data points for 40 computer science categories, chosen proportionally based on the labels in the public split.

*3.1.2 Baselines.* We compare our model with several pre-trained transformer-based language models. In the zero-shot scenario, we calculate the similarity between the condensed node description and the labeled text with a prompt, and predict the most similar class.

• BERT [5] and BERT*: BERT is a pre-trained transformer model that employs masked language modeling as its pre-training strategy. In the pre-training phase, BERT generates detailed bidirectional representations by incorporating both left and right context across all layers from unlabeled text. BERT*, on the other hand, is a model

refined through fine-tuning the BERT model using masked language modeling with text from a specific dataset. This process enhances the model's understanding and enriches its knowledge of the downstream datasets.

• RoBERTa [18] and RoBERTa*: RoBERTa is a variant of the BERT model which incorporates additional training techniques and data augmentation strategies to improve the model's performance further. Similar to BERT*, we finetune RoBERTa to obtain RoBERTa*. To further examine the effectiveness of our *GraphTranslator* aligning GM to LLM, we compare with the methods only using text to query LLM:

• LLM+$s_v$: It directly appends the original attribute description $s_v$ to instruction, serving as input for ChatGLM2-6B.

• LLM+$s_v$+$s_{\mathcal{N}(v)}$: It simply merges the vanilla text attribute of node and neighbors to instruction to serve as input for ChatGLM2-6B.

*3.1.3 Model Details.*

**Graph model.** In the Taobao dataset, we utilize GraphSAGE [9] to aggregate neighbors' information and generate node embeddings. The GraphSAGE model employs a 2-layer aggregation and samples 10 neighbors for each layer. The dimensions of the input, intermediate, and output layers are set to 768. For the ArXiv dataset, we configure the intermediate dimension as 1024, while maintaining the remaining settings identical to the Taobao dataset.

**Large Language Model.** The LLM model employed in this research is ChatGLM2-6B, which possesses a total parameter size of approximately 6 billion. The model is composed of 28 Transformer blocks, each with a hidden size of 4096 and 32 attention heads. Additionally, the feed-forward network incorporates an intermediate layer dimension of 13,696. The vocabulary size is set at 65,024, while the maximum sequence length is capped at 32,768.

**Translator.** The Translator module is implemented based on a BERT model, BERT-base-Chinese for the Taobao dataset and BERT-base-English for the ArXiv dataset. It includes 12 layers of Transformer blocks, with alternate layers conducting cross-attention between node embeddings and query tokens, which includes 32 tokens with the dimension of 768. The attention head's number is set to 12. The maximum sequence length is set at 512, with a vocabulary size of 21,128 for the Taobao dataset and 30,522 for the ArXiv dataset. The final output of the Translator contains 32 embeddings with the dimension of 768.

*3.1.4 Experiment Environment.* All experiments are conducted on a Linux server with four GPU (Tesla V100, Memory 32G) and CPU (Intel(R) Xeon(R) Platinum 8163 CPU @ 2.50GHz),and its operating system is Ubuntu 20.04. We implement the proposed GraphTranslator with deep learning library PyTorch and PyTorch Geometric. The Python and PyTorch versions are 3.8 and 1.12, respectively.

## 3.2 Zero-shot Node Classification

Zero-shot classification, as an emergent capability of LLMs, allows the model to predict the class that has not been seen during the training phase. Our *GraphTranslator* , aligned to LLMs, is expected to classify the nodes to unseen classes. We conduct zero-shot node classification on four tasks as follows: (1) Taobao Lifestage prediction aims to reason the life stage of user to three categories [Single, Married, Parented]. (2) Taobao Cat Owner prediction is to infer

**Table 1: Results on zero-shot node classification.**

| Dataset | Metric | BERT | RoBERTa | BERT* | RoBERTa* | LLM+$s_v$ | LLM+$s_v$+$s_{\mathcal{N}(v)}$ | GraphTranslator |
|---|---|---|---|---|---|---|---|---|
| Taobao (Lifestage) | Legality Rate (%) | 100.00 | 100.00 | 100.00 | 100.00 | 50.10 | 55.57 | 58.80 |
| | Accuracy (%) | 34.73 | 33.10 | 32.97 | 34.53 | 33.46 | 34.59 | **35.33** |
| | Recall (%) | 34.73 | 33.10 | 32.97 | 34.53 | 33.46 | 34.59 | **35.33** |
| | Macro-F1 (%) | 27.17 | 24.56 | 25.06 | 25.73 | 31.63 | 32.60 | **32.62** |
| Taobao (Cat Owner) | Legality Rate (%) | 100.00 | 100.00 | 100.00 | 100.00 | 31.20 | 45.43 | 98.97 |
| | Accuracy (%) | 51.13 | 50.87 | 49.03 | 48.77 | 51.92 | **58.55** | 50.99 |
| | Recall (%) | 87.40 | 60.40 | 63.27 | 11.73 | 12.82 | 45.56 | **95.69** |
| | Macro-F1 (%) | 43.73 | 50.42 | 47.98 | 40.62 | 21.05 | 52.96 | **66.14** |
| Taobao (Vehicle Owner) | Legality Rate (%) | 100.00 | 100.00 | 100.00 | 100.00 | 63.97 | 86.17 | 94.60 |
| | Accuracy (%) | 47.53 | 47.93 | 47.37 | 48.73 | 46.74 | 49.09 | **49.40** |
| | Recall (%) | 59.00 | 54.73 | 51.53 | 64.60 | 63.01 | 61.29 | **83.27** |
| | Macro-F1 (%) | 46.83 | 47.69 | 47.28 | 47.41 | 54.62 | 55.15 | **61.87** |
| ArXiv | Legality Rate(%) | 100.00 | 100.00 | 100.00 | 100.00 | 99.15 | 99.40 | 97.8 |
| | Top-1 Acc (%) | 1.63 | 3.55 | 14.53 | 6.95 | 14.07 | 17.90 | **28.48** |
| | Top-3 Acc (%) | 7.63 | 11.98 | 29.60 | 16.53 | 26.98 | 28.43 | **37.62** |
| | Top-5 Acc (%) | 28.00 | 22.93 | 38.30 | 23.75 | **42.46** | 37.99 | 39.87 |

whether the user has cat. (3) Taobao Vehicle Owner prediction is to infer whether the user has vehicle. (4) Arixv CS sub-catgories prediction is to determine the paper's category from 40 sub-categories.

The effectiveness is evaluated by utilizing several metrics: (1) Legality Rate, follow [36], we use legality rate to measure the ratio that model produces valid answers. (2) Accuracy stands for the percentage of the correct predictions over total predicted samples. (3) Recall is calculated as the positive class recall rate in the 2 class task and macro-recall rate in the muli-class task. (4) F1-score is defined as the harmonic mean of precision and recall. The higher F1-score and legality rate hint the better effectiveness. (5) Top-k classification accuracy represents the percentage of the correct label is among the top-k predicted labels which is only adopted by the Arxiv sub-category prediction task. Note that the LLM-based models typically make predictions within the format of text rather than discrete labels, so we employ regular expression matching to extract the predicted class from response for evaluation. More details can be found in the Appendix C.

The experimental results on Taobao and ArXiv are presented in Table 1, where we have the following observations:

• Our model *GraphTranslator* achieves better performance than most of baselines, which indicates GM can greatly benefit from LLMs within our *GraphTranslator*. The BERT-based method performs poorly, as it relies solely on similarity calculations and is unable to handle complex zero-shot scenarios. Our *GraphTranslator* model performs better than *Vanilla LLM*, including LLM+$s_v$ and LLM+$s_v$+$s_{\mathcal{N}(v)}$, since LLM directly processes the raw text that contains both node and neighbor attribute, bringing noises and excessive complexity. It demonstrates the superior of *GraphTranslator* to extract the graph information using the soft prompt translated from node embedding.

• In LLM-based methods, our approach achieves the highest legality rate. The reason may be that LLM+$s_v$ only inputs the raw text attribute $s_v$ of nodes into the LLM, which contains limited information and poses significant reasoning challenges for the LLM. Then LLM+$s_v$+$s_{\mathcal{N}(v)}$ enriches it with the attributes of neighbors

$\mathcal{N}(v)$, and gain the higher legality rate but still suffer from the noisy and redundant text. Especially for Taobao dataset, it is challenging for LLM to derive answers of instructions from the intricate shopping history within user attribute. While our *GraphTranslator* introduces a Producer module to succinctly summarize $s_v$, $s_{\mathcal{N}(v)}$ and its commonalities, providing rich content while reducing noise. And *GraphTranslator* takes node representations translated by the Translator, serving as a soft graph prompt that can encapsulate more intricate details than discrete text. Moreover, *GraphTranslator* projects input embeddings into fixed-length tokens, facilitating the comprehension of LLM for graph information.

• Our *GraphTranslator* exhibits a particularly notable improvement in recall for positive instances. Taking the Cat Owner prediction on the Taobao dataset as an example, LLM+$s_v$+$s_{\mathcal{N}(v)}$ method requires explicit product terms in the text, such as "cat food" or "cat litter", to predict as a positive instance. However, the translated node embedding in *GraphTranslator* encodes and compresses the information, where the products implicitly related to cats are also summarized as cat product explicitly. Hence, *GraphTranslator* is more adept at accurately identifying the positive instances, which is crucial for industrial applications.

## 3.3 Graph Question Answering (GQA)

To further reveal the potential and commercial value of our *Graph-Translator* across a wide range of open-ended applications, we showcase the graph question answering (GQA) experiments on Taobao dataset. We query the LLM in a multi-turn dialogue format to deeply investigate the capability of our *GraphTranslator* to extract, explain and reason the unseen node embedding.

To provide the quantitative analysis of *GraphTranslator*, we build an evaluation set by randomly sampling 100 nodes and constructing three questions as follows: (1) User Understanding: "Please summarize the user's interests." (2)Friends Understanding: "Please summarize the common interests and preference of these friends." (3)Friendship Analysis: "Why does this user become friends with

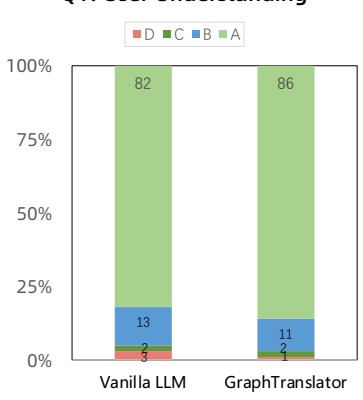
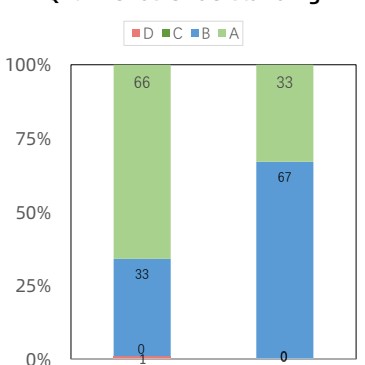
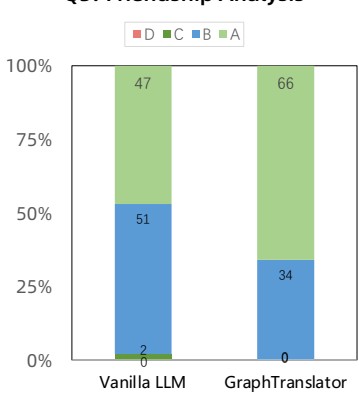

**Figure 3: Quantitative analysis of graph question answering. The order of response quality ranking is as follows: A > B > C > D.**

these people? " For *GraphTranslator* the translated user embedding is only concatenated with the first question, serving a soft prompt. As a comparison, we directly feed the text attributes of user and neighbors to ChatGLM2-6B. More details of prompts can be found in the Appendix C. As questions are open-ended, we employ ChatGPT(GPT-3.5-turbo-16k) as the evaluator to perform quantitative analysis. Following [29], we gather question-answering pairs of each test sample and use the four-level rating system:

- Rating-A: The answer is correct and concise, the information is correct, and the reasoning is accurate.
- Rating-B: The answer is reasonable, with minor errors or imperfections.
- Rating-C: The answer is relevant to the question, but has obvious errors or inaccuracies in the content.
- Rating-D: The response is irrelevant or completely invalid.

The comparison results are presented in Figure 3, where we have following observations:

• Only provided unseen node embedding as prompt for LLM, our *GraphTranslator* gets 185 A in total, showcasing strong performance comparable to *Vanilla LLM* which gets 195. This is because that *GraphTranslator* are trained on the low-noise and low-pass descriptions generated by our Producer, thus the trained Translator can extract high-quality information from the node embedding for multi-turn dialogue.

• One can observe that our *GraphTranslator* achieves better performance in Q1 and Q3, especially for the challenging question Q3, which requires model to understand graph and reason why these people become friends. It demonstrates the superior of *GraphTranslator* to extract graph information based on node embedding.

A detailed case is showed in Figure 4. We analyse several basic abilities which are reflected in the model's response, and have the following observations:

• Graph understanding: Taking the summarization of user's interests in Q1 and friends' interests in Q2 as examples, the correct responses require the model to recognize that both the user and friends are interested in cars. The *Vanilla LLM* can understand the instructions, but struggles to extract the key characteristics. Specifically, in response A1, *Vanilla LLM* lists numerous specific

products as marked by the red strikethrough, which belong to friends, not users. And by A2, *Vanilla LLM* summaries too many hobbies, but only car is the true shared interest. The reason is that the lengthy prompt in Q1 (over 1000 words) may disturb the attention mechanism of LLM. On the other hand, as highlighted in green, our *GraphTranslator* identifies the primary interests and hobbies in A1, analyses the related personality and life needs, and in A2, successfully concludes that their common interest is cars. It demonstrates the superior ability of *GraphTranslator* to extract and interpret graph information using the soft prompt translated from node embedding.

• Reasoning ability: The question Q3 requires models to understand graph information and reason why these people become friends. The *Vanilla LLM* captures too many noise information of friends, thus tend to give a general explanation. Our *GraphTranslator* provide explanations from three perspectives, i.e., the similarity of hobbies, the maintenance of friendship, and friend influence, which are all centered around the user and friends' shared interest of cars. The final answer is fundamentally accurate, and it presents an explicit and logical reasoning process.

• Multi-turn dialogue ability: By providing graph information only in the first question Q1, we can observe the improvement in our model's multi-turn dialogue capability compared to *Vanilla LLM*. Our *GraphTranslator* maintains a consistent graph-centric response throughout the conversation. This capability is attributed to the low-noise and low-pass descriptions generated by our Producer, thus the trained Translator can extract high-quality information from the node embedding for multi-turn dialogue.

## 4 DISCUSSION

In this paper, we investigate the alignment of graph models (GMs) with Large Language Models (LLMs) to augment the capability of GMs in addressing open-ended tasks. While the preliminary experiments reveal our *GraphTranslator* potential on open-ended applications, there are several limitations listed as follows:

• The Producer plays a pivotal role in dictating the overall quality of the Translator model. Given the low-pass property of most GNNs, our Producer leverages LLMs to summarize node, neighbors, and

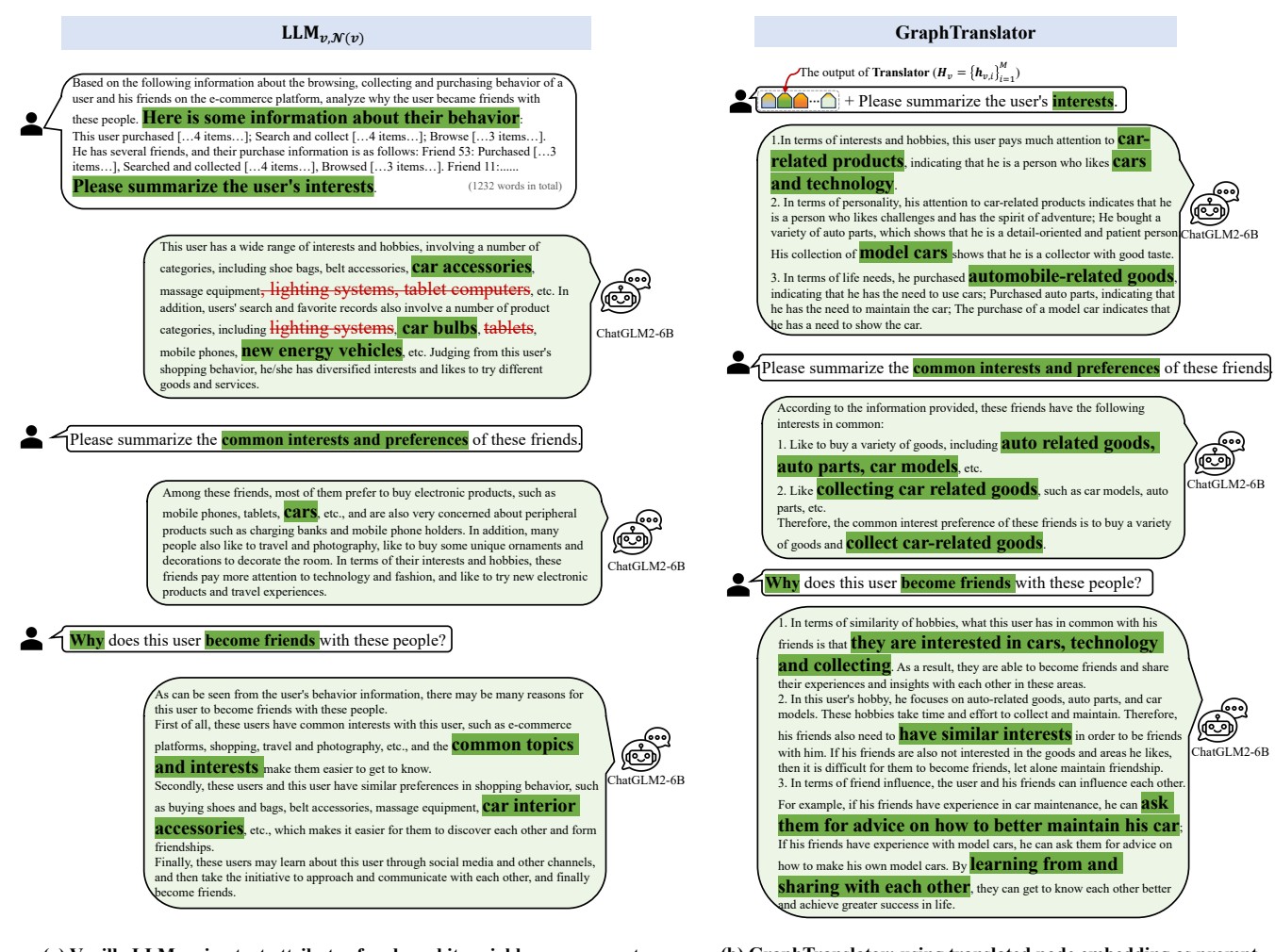

Figure 4: A case of graph question answering on Taobao Dataset.

their commonalities. In future work, more topology information encoded in node embeddings, like node degree, can be included in the description in order to reduce information loss. Moreover, due to limited resources, we employed ChatGLM2-6B to generate descriptions in Producer. In future work, we can use larger-scale LLMs such as ChatGPT, to improve the quality of the generated text descriptions. And integrating novel LLM utilities and techniques, such as with Chain-of-Thought [32] and AutoPrompt [24], also could further improve the performance.

• In the experiment, we only have labels for quantitative analysis in our zero-shot node classification, and for the GQA task, we merely showcase our *GraphTranslator* performance through specific cases. To offer a complete and quantitative evaluation of model capabilities in open-ended tasks, such as graph understanding, explaining, reasoning, and multi-round conversation, it's important to develop an evaluation dataset and devise corresponding metrics in future work.

## 5 CONCLUSIONS

In this paper, we propose a novel framework to align graph models (GMs) to LLM, named *GraphTranslator*, aiming to utilize the extended interface of LLMs to offer various open-ended tasks for GM. *GraphTranslator* introduces a Translator module to eliminate the modality gap, by converting node embeddings learned by GM to a set of tokens. For further training, a Producer module is designed to generate the alignment data, through seamlessly textualizing the information encoded in node embeddings. We evaluate our method on real-world datasets for open-ended tasks. The experimental results demonstrate the effectiveness of *GraphTranslator* on zero-shot node classification. The preliminary graph question answering experiments indicate the capability of our *GraphTranslator* to extract, explain and reason the graph information, revealing the potential and commercial value.

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

# A  RELATED WORK

## A.1  Graph Representation Learning

Graph representation learning, including earlier shallow graph embedding [23] and graph neural networks [9, 15, 27], aims to capture and encode these relationships in a way that facilitates various downstream tasks. Traditional graph models primarily focus on supervised learning, requiring a substantial amount of labeled data for tasks such as node classification. These methods achieve strong performance when sufficient labeled data is available, but tend to underperform in scenarios with limited labeled data. Inspired by pre-trained language models, the new paradigm of "Pre-train, Prompt, and Predict" has been recognized for its effectiveness in addressing few-shot downstream tasks [7, 14, 20, 25, 26, 40, 43]. Despite their advancements, achieving zero-shot learning and open-ended tasks remains challenging.

## A.2  Large Language Model for Graph

The NLP landscape has recently been revolutionized by language modeling (LM), which is one of the major approaches to advancing the language intelligence of machines [5]. The goal of LM is to model the generative likelihood of text for predicting future (or masked) token probabilities. Recently, researchers have observed that when the model sizes of the pre-trained LMs up to a certain scale, the LMs will showcase some remarkable capabilities, named emergent abilities [30]. Here we use the term "large language models" (LLMs) to denote such language models that have the extensive number of billions parameters, and have been pre-trained on vast corpora of data [1]. These LLMs, like ChatGPT and GPT4 [22], can effectively follow language even multi-modal instructions, aligned with human intent to complete various real-world tasks. More recently, applying LLMs for graph domain has received several preliminary research experimental trials already, which can be categorized into two classes [3]: The first, LLMs-as-Enhancers [4, 6, 11], augments node text attributes with LLM knowledge, and still utilizes GMs for predictions. They can produce accurate predictions on traditional tasks, but fail to make full use of the capabilities of text generation, instruction following for open-ended tasks. The second, LLMs-as-Predictors, treats nodes as tokens or text and uses LLMs as the standalone predictor, which often can generate imaginative content but training/inferring LLMs can be time-consuming. This can actually hinder their widespread deployment for web-scale pre-defined tasks.

# B  EXPERIMENT DETAILS

## B.1  Training Details.

● Pre-training Graph Model Phase. In the pre-training phase, we employ link prediction as the self-supervised task for pre-training the graph model. For each batch in the Taobao dataset, we randomly select 1024 positive and negative edges. In the case of the ArXiv dataset, we sample 65,536 edges for each batch, and the ratio of positive and negative examples is set to 1:1 in both datasets. We use Adam as the optimizer and set the learning rate and weight decay to 1e-4 and 1e-3 for the Taobao dataset. For the Arxiv dataset, we set the learning rate to 0.01 and omit the weight decay.

● Pre-training LLM Phase. We adopt the well-pretrained ChatGLM2-6B [2], which is the second-generation version of the open-source bilingual (Chinese-English) chat model ChatGLM-6B.

● Stage 1 Training Phase. We utilize the 274,168 and 90,941 pairs of node representation and textual description for Taobao and ArXiv datasets respectively. We employ a batch size of 16 and train the *GraphTranslator* for 8 epochs. We adopt the AdamW optimizer and set the learning rate and weight decay to 1e-6 and 0.05 respectively. Furthermore, to accelerate the training process and enlarge the batch size of gradient backpropagation, we incorporate the technique of *gradient accumulation* with 32 steps. The implementation of *gradient accumulation* results in an augmented batch size of 512 for gradient backpropagation.

● Stage 2 Training Phase. We maintain the same number of pairs as in stage 1. Due to memory limitations, we employ a batch size of 8 and train the *GraphTranslator* for 3 epochs. We adopt the AdamW optimizer, and set the learning rate and weight decay to 1e-6 and 0.05 respectively. We also incorporate *gradient accumulation* technique during the Stage 2, which results in an augmented batch size of 256 for gradient backpropagation.

## B.2  Further Experiment

To validate the effectiveness of training strategies, we compare our *GraphTranslator* with its variants, "Stage 1 Only" and "Stage 2 Only". Taking Arxiv dataset as an example, the results are presented in the Table 2. We observe that the training of stage 1, despite effectively aligning graph embeddings and texts, fails to map graph embeddings into the semantic space of the LLM. As a result, the LLM is hard to understand semantic information and it exhibits significantly lower performance compared to *GraphTranslator*. On the other hand, although stage 2 bridges the gap between graph embedding and LLM directly, it lacks the understanding between embeddings and texts, thus contributing to the sub-optimal performance. In conclusion, the two-phase training of *GraphTranslator* is crucial for enabling LLM to comprehend the graph information, ultimately leading to the optimal results.

# C  PROMPT DESIGN

The prompts of Taobao and ArXiv datasets are presented in the Table 3, 4, 5, 6. For the prompts of the Taobao dataset, we translate them into English with Youdao [3]. The prompts for the inference stage are shown in Table 5, we force the model to answer the number corresponding to the label in the Taobao dataset. Therefore, we employ regular expression matching to identify the relevant labels based on the digital number. For the ArXiv dataset, we extract labels based on the textual descriptions associated with each category.

---

[2]https://huggingface.co/THUDM/chatglm2-6b
[3]https://fanyi.youdao.com/

**Table 2: Impact of Stage 1 and Stage 2**

|  | Legality Rate | Top-1 | Top-3 | Top-5 |
|---|---|---|---|---|
| GraphTranslator (Stage1 Only) | 98.28 | 8.22 | 16.94 | 25.92 |
| GraphTranslator (Stage2 Only) | 99.60 | 16.94 | 27.54 | **41.24** |
| GraphTranslator | 97.80 | **28.28** | **37.63** | 39.88 |

**Table 3: Prompts For the Producer**

| Dataset | Step | Prompt |
|---|---|---|
| Taobao | User behavior summary | User Behavior Description: <User Behavior Description>. Please summarize the characteristics of this user according to the product behavior information. The answer format is: What kind of characteristics does the user have in terms of interests, hobbies, personality traits, and life needs |
|  | Neighbor behavior summary | Neighbor Behavior Description: <Neighbor Behavior Description>. Please summarize most of the similarities that this user's friends have based on the product behavior information. The answer format is: What do several friends of this user have in common in interests, hobbies, personality traits, and life needs? |
| ArXiv | Paper Summary | The title and abstract of this paper are as follows: <Title text> \t <Abstract text>. please summarize this paper and list five keywords of this paper. |
|  | Neighbor Paper Summary | The paper title and abstract are provided as follows: <Title text> \t <Abstract text>. \n <Title text> \t <Abstract text>.... \n Please summarize the topic and content of these papers. All answers are in English and No Chinese in your answer |

**Table 4: Prompts For the Training Stage**

| Dataset | Prompt |
|---|---|
| Taobao | Based on the product information, please describe the characteristics of this user, and the common characteristics of his friends in interests, hobbies, personality traits, and life needs. |
| ArXiv | Please summarize the topic and content of the paper and its cited papers in English. |

**Table 5: Prompts For the Inference Stage**

| Dataset | Step | Prompt |
|---------|------|--------|
| Taobao | Lifestage Prediction | Question: You are a home marketing analyst, you must use only one character to answer the guess about the user's family situation based on the user's purchase information for yourself or for the family, the answer format is [X] : Answer if the user is a single person who is not in a relationship [1], answer if the user is a childless person who is in a relationship and has no children [2], answer if the user is a married and childless person who has children in the family [3]. Answer: |
|  | Cat Owner Prediction | Question: You are a brand of cat food operator, and you need to distribute the brand of cat food experience kits to users who have pets at home. According to the user's product information, you must use only one character to answer whether to issue brand cat food experience to this user, the answer format is [X]: If it is not necessary to issue experience cat food to this user answer [0], if issue experience cat food to this user answer [1]. Answer: |
|  | Vehicle Owner Prediction | Question: You are an automobile traffic safety propagandist of the government transportation department, and you need to spend time and energy to popularize automobile driving safety education for users with cars. Please use only one character to answer whether it is necessary to conduct automobile driving safety education for this user according to the product information of the user. The answer format is [X]: If the user does not have a car at home and therefore does not require a car driving safety education answer [0], if the user has a car at home requires a car driving safety education answer [1]. Answer: |
| ArXiv | CS sub-categories Prediction | The summary of the paper is as follows: <Paper Summary>. The summary of the related paper is as follows: <Neighbor Paper Summary>. \n Question: Based on the summary of the above paper and citations, please determine into which of the following 40 ArXiv CS sub-categories would this paper most likely fall? categories: <Artificial Intelligence; Hardware Architecture; Computational Complexity; Computational Engineering, Finance, and Science; Computational Geometry; Computation and Language; Cryptography and Security; Computer Vision and Pattern Recognition; Computers and Society; Databases; Distributed, Parallel, and Cluster Computing; Digital Libraries; Discrete Mathematics; Data Structures and Algorithms; Emerging Technologies; Formal Languages and Automata Theory; General Literature; Graphics; Computer Science and Game Theory; Human-Computer Interaction; Information Retrieval; Information Theory; Machine Learning; Logic in Computer Science; Multiagent Systems; Multimedia; Mathematical Software; Numerical Analysis; Neural and Evolutionary Computing; Networking and Internet Architecture; Other Computer Science; Operating Systems; Performance; Programming Languages; Robotics; Symbolic Computation; Sound; Software Engineering; Social and Information Networks; Systems and Control>. Please give 5 likely categories, in order from most likely to least likely, and give your reasoning. Provide response in JSON format with the following keys: category, reason. |

## Table 6: Prompt Design For the GQA

|  | Vanilla LLM | GraphTranslator |
|---|---|---|
| Question1 | Please conduct some theoretical analysis of social impact based on the browsing, collecting, and purchasing behaviors of a user and his or her friends on an e-commerce platform. Here is some information about their behavior: {user_and_friends_behaviors} Please summarize the interests of this user. | <Embedding>. Please summarize the user's interests |
| Question2 | Please summarize the common interests and preferences of these friends | Please summarize the common interests and preferences of these friends |
| Question3 | Why does this user become friends with these people? | Why does this user become friends with these people? |
| Evaluator(For ChatGPT) | The following is a 3-round dialogue. Please judge the quality of the responses based on the given criteria. The criteria are: A: The answers are correct and concise, the information is completely correct, and the reasoning is accurate. B: The answers are reasonable, with a very small number of information errors or imperfections. C: The answer is relevant to the question, but has obvious errors or inaccuracies in the content. D: The response is not relevant or completely invalid Here's the conversation: [Round 1]Q:... A:... [Round 2]Q:... A:... [Round 3]Q:... A:... Please judge each answer one by one, output the rating results of 3 questions, and return in the form of a list. Format reference: [C, B, B] | The following is a 3-round dialogue. Please judge the quality of the responses based on the given criteria. The criteria are: A: The answers are correct and concise, the information is completely correct, and the reasoning is accurate. B: The answers are reasonable, with a very small number of information errors or imperfections. C: The answer is relevant to the question, but has obvious errors or inaccuracies in the content. D: The response is not relevant or completely invalid Here's the conversation: Background: {user_and_friends_behaviors} [Round 1]Q:... A:... [Round 2]Q:... A:... [Round 3]Q:... A:... Please judge each answer one by one, output the rating results of 3 questions, and return in the form of a list. Format reference: [C, B, B] |

