# OpenReview forum: "GraphTranslator: Aligning Graph Model to Large Language Model for Open-ended Tasks"
_ACM.org/TheWebConf/2024/Conference — TheWebConf24 Oral_

### Official Review · Reviewer_z64r · 2023-11-15

**Novelty:** 6
**Technical Quality:** 6

**Review:**

The paper titled 'GraphTranslator: Aligning Graph Model to Large Language Model for Open-ended Tasks' explores the alignment of large language models (LLMs) and graph models (GMs), presenting a novel approach named GraphTranslator to overcome limitations in handling both pre-defined and open-ended tasks within the graph domain. The paper provides a forward-looking perspective on addressing existing limitations and expanding the capabilities of graph models.

## Reasons to Accept
**Clear and Comprehensive Motivation**
The motivation behind the research is articulated clearly and comprehensively. While existing GMs primarily focus on pre-defined tasks, the paper recognizes the gap in the literature—where current methods cannot handle pre-defined and open-ended tasks simultaneously. The introduction of GraphTranslator is a novel attempt to align GMs and LLMs to bridge the gap.

**Theoretical Soundness**
The theoretical foundation of the proposed method is robust. In the introduction, the paper succinctly outlines two major challenges in the graph domain—modality gap and the lack of alignment data. The innovative GraphTranslator is designed to tackle these challenges through its Translator module, addressing the modality gap, and the Producer module, handling the scarcity of alignment data.

**Effective Visual Representation**
The paper excels in visual representation, particularly in Figure 2. The graphical illustrations enhance the clarity of the proposed GraphTranslator's architecture and its functioning. The visual aids contribute significantly to the reader's understanding of the method.

**Clarity and Coherence**
The paper is well-organized and the ideas are presented with clarity and coherence. The logical flow of information from the motivation to the proposed solution enhances the overall readability of the manuscript.


## Reasons to Reject
**Lack of Performance Comparison**
The introduction claims that existing methods suffer from issues of being slow and costly, but the experimental results do not explicitly demonstrate how the proposed method excels in terms of speed and cost efficiency. Without a comparative analysis showcasing the advantages of the proposed approach, it remains unclear how the new method addresses the purported shortcomings of existing solutions.

**Concerns Regarding Producer Functionality**
The Producer's role is to construct alignment data in the form of (node embedding, textual description) pairs, where the textual descriptions are generated by LLMs. The accuracy of these textual descriptions is critical, yet the paper does not sufficiently address how the correctness of the LLM-generated descriptions is ensured. Existing literature suggests that LLMs might not possess a complete and accurate understanding of graphs, potentially leading to inaccuracies in the generated descriptions. The paper should provide a robust explanation or validation mechanism to address this concern.

References:
1. GPT4Graph: Can Large Language Models Understand Graph Structured Data? An Empirical Evaluation and Benchmarking https://arxiv.org/pdf/2305.15066.pdf
2. EVALUATING LARGE LANGUAGE MODELS ON GRAPHS: PERFORMANCE INSIGHTS AND COMPARATIVE ANALYSIS https://arxiv.org/pdf/2308.11224.pdf

**Limited Exploration of Input Settings**
The paper primarily employs LLMs with input settings resembling zero-shot prompts. However, existing literature has explored a broader range of input configurations, including zero-shot prompts, few-shot prompts, zero-shot prompts with CoT, and few-shot prompts with CoT. Exploring additional input settings like few-shot prompts should not inherently conflict with the chosen zero-shot node classification task.

References:
1. Exploring the Potential of Large Language Models (LLMs) in Learning on Graphs https://arxiv.org/pdf/2307.03393.pdf
2. EVALUATING LARGE LANGUAGE MODELS ON GRAPHS: PERFORMANCE INSIGHTS AND COMPARATIVE ANALYSIS https://arxiv.org/pdf/2308.11224.pdf
3. CAN LLMS EFFECTIVELY LEVERAGE GRAPH STRUCTURAL INFORMATION: WHEN AND WHY https://arxiv.org/pdf/2309.16595.pdf

**Reliance on ChatGPT for Evaluation**
The paper relies on ChatGPT as an evaluator for the results. However, existing studies have highlighted issues with using ChatGPT or similar models as evaluators due to concerns such as position bias. The paper should acknowledge these limitations and can consider incorporating human evaluation to provide a more reliable assessment of the proposed method.

References:
1. Judging LLM-as-a-Judge with MT-Bench and Chatbot Arena https://arxiv.org/pdf/2306.05685.pdf

**Limited Diversity in Model Selection**
The paper exclusively employs GraphSAGE as the GM and ChatGLM-6B as the LLM. Existing literature explores a broader spectrum of GMs and LLMs, facilitating comparative analyses. The paper should consider incorporating a variety of GMs and LLMs into the experiments to ensure a comprehensive evaluation. Failing to do so may weaken the generalizability and robustness of the proposed method. If the choice is intentional, the paper should provide a robust justification for limiting the experimentation to GraphSAGE and ChatGLM-6B.

References:
1. Exploring the Potential of Large Language Models (LLMs) in Learning on Graphs https://arxiv.org/pdf/2307.03393.pdf

**Questions:**

## Questions
**Task Categorization and Solution Application**
The paper presents distinct solutions for pre-defined and open-ended tasks, which raises the question of practical implementation. In real-world problem-solving scenarios, is it necessary to determine the task category before applying the corresponding method? If so, how is this categorization achieved? Is it a manual judgment, or are there automated criteria to distinguish between task types?

**Soft Prompt in LLM Input**
The paper introduces soft prompts in the input of LLMs, which include information representing graph features. Is the length of soft prompt fixed, or does it vary based on the complexity of the graph? If there is variability, how does the system determine the optimal length of the soft prompt? Additionally, in cases where the graph is highly complex, and the soft prompt becomes extensive, how is the potential impact on the model's performance managed? Furthermore, if the length of the soft prompt exceeds the input length limitations of LLMs, how is this handled, and what implications does it have on the overall effectiveness of the approach?

**Reviewer Confidence:**

3: The reviewer is confident but not certain that the evaluation is correct

**Scope:**

4: The work is relevant to the Web and to the track, and is of broad interest to the community

---

### Official Review · Reviewer_oo6h · 2023-11-22

**Novelty:** 4
**Technical Quality:** 5

**Review:**

This paper is about bridging the gap between pretrained graph models (GMs) and large language models (LLMs) for handling both pre-defined tasks and open-ended tasks. It introduces a novel framework named GraphTranslator, which uses a Translator module to convert node embeddings generated by GMs into a language that LLMs can interpret, thus allowing GMs to leverage the flexible instruction-following capabilities of LLMs.

Strengths:
1.	The paper is overall well-written and easy to follow.
2.	The proposed GraphTranslator model provides a new way to integrate graph models with large language models, enabling the handling of both structured pre-defined tasks and flexible open-ended tasks in a unified manner.
3.	They introduce a Translator module within the GraphTranslator framework, coupled with a Producer module to generate necessary alignment data, which facilitates the translation of node embeddings into a language format interpretable by LLMs, thereby enhancing performance on tasks such as zero-shot node classification and graph question answering.

Weaknesses:
1.	Although the proposed model is relatively new in the Graph Learning community, its overall architecture is quite similar to that of InstructBLIP, which essentially replaces the frozen image encoder in InstructBLIP with the frozen Graph Model in this application scenario. Hence, I feel the technical contribution here is not that significant.
2.	The GraphTranslator model appears to be a complex system that integrates two sophisticated types of models: graph models and large language models. Such complexity could result in high computational demands, potentially limiting the scalability of the model when applied to very large datasets or complex graph structures. There could be concerns regarding the time and resources needed to train and fine-tune the system, as well as the compute power required for inference, especially in resource-constrained environments.
3.	The performance of the GraphTranslator heavily relies on the quality of the alignment data produced by the Producer module. This data is crucial for the model to effectively translate node embeddings into textual tokens that LLMs can process. If the alignment data is not of high quality, which means it does not capture the nuanced relationships between graph nodes and their textual descriptions accurately, the model might produce suboptimal results. The process of creating this alignment data is not trivial and could be prone to errors, which in turn would affect the model's performance.
4.	The paper attempts to use LLMs to enhance the interpretability of GMs for open-ended tasks. However, the extent to which the model's predictions are interpretable and explainable to end-users is not fully explored. It is crucial for models not only to perform well but also to provide insights into their decision-making processes, especially in domains that require a high degree of trust and understanding from the users.

**Questions:**

It seems that the overall architecture of the proposed method is quite similar to that of InstructBLIP, could you point out the key difference (in other words, the key technical contribution of the proposed model)?

**Reviewer Confidence:**

3: The reviewer is confident but not certain that the evaluation is correct

**Scope:**

4: The work is relevant to the Web and to the track, and is of broad interest to the community

---

### Official Review · Reviewer_ywya · 2023-11-22

**Novelty:** 4
**Technical Quality:** 4

**Review:**

This interesting paper attempts to combine nodal representation with text attributes associated with the nodes in conjunction with a pre-trained LLM to address zero shot node classification and graph question answering tasks.

Section 2.7 on model training, a critical section, is poorly written. For example, (in Stage 1), the authors state that they compute cross-entropy loss on the text "Now we replace the [CLS] token with [DEC] token for the generation task. By optimizing the cross entropy loss between the generated text and the actual description $t_v$ , the $\pmb{Q}$ is forced to capture more details in $z_v$ related to the $t_v$." While this reviewer can guess, it would have been better to see the mathematical formulation of the loss function. The descriptions for stage 2 are even more sparse; while there are several ways to accomplish stage 2, the authors must provide more specific details on their training regimen to ensure reproducibility.

The two datasets used in the evaluation are quite different. In particular, this reviewer found it curious that the nodal attributes for the Taobao dataset include " user behaviors like purchases, searches, browsing, favorites, and cart additions. " (line 498). This is strikingly different from the ArXiv dataset. While the dataset diversity is excellent, the question of graph node, to node text (and neighborhood text) alignment is more challenging in the Taobao dataset. It is unclear how the proposed work (esp. the section on training) addresses this issue.

The baseline zero-knowledge tasks seem quite arbitrary. Why select these tasks (e.g., cat ownership) and not others (e.g., educational attainment; income level) that would seem more appropriate to an e-commerce website? The authors should provide a more detailed justification for selecting these tasks.

The evaluation of the graph question answering tasks seems to be problematic. Why did the authors pick these tasks? The third task “Why does this user become friends with these people?" seems very difficult. The last task is well known to be impossible with only observational data since we cannot disentangle latent homophily and socialization effects from only observational data. For example, Fig 4(b) shows that the proposed model indicates that "they are interested in cars, technology and collecting" as a reason for friendship. Could one friend have influenced the other to be interested in these topics? Or, could it be that the two friends were interested in these topics, and that is why they became friends? The authors should provide a more detailed justification for selecting these tasks. Second, using ChatGPT to evaluate this task needs justification; why not use human subject evaluation as a gold standard? ChatGPT model's errors may correlate with the mistakes of the proposed model, and some baseline analysis is needed to justify this choice. Further, using reference [29] to justify a rubric for evaluating this task seems odd since [29] deals with automatic prompt instruction generation.

**Questions:**

* I might have missed the reasoning, but Table 1 indicates that the legality rate of the BERT-based models is the highest (100%). What might be the reason for this?
* Please justify the choice of the zero-shot tasks and the graph question-answering tasks in this paper.

**Ethics Review Description:**

no concerns

**Reviewer Confidence:**

3: The reviewer is confident but not certain that the evaluation is correct

**Scope:**

4: The work is relevant to the Web and to the track, and is of broad interest to the community

---

### Decision · Program_Chairs · 2024-01-22

**Decision:**

Accept (Oral)

**Comment:**

The paper studies a substantial problem, the alignment of LLMs with graph data. It proposes a novel attention-based adapter module to bridge the pre-trained LLM and GM, by projecting node embeddings from the GM to prefix tokens of LLMs. Despite many concerns about choices of tasks and evaluation settings have been raised by reviewers, they all held positive opinions of the paper in general.

 Personally, I really appreciated the tasks presented in this paper, and applying GM and LLM to GQA and open-ended tasks was a very bold and interesting attempt. Although, as reviewers have mentioned, these tasks and settings require more rigid judgments and discussions, but overall I enjoyed it very much.

 To summarize, although the paper still needs to be polished with respect to experimental settings, I think the novel idea of this paper will benefit the graph research community a lot. Therefore, my final recommendation is Accept.